# Quality Measures in Systemic Sclerosis

**DOI:** 10.3390/diagnostics13040579

**Published:** 2023-02-04

**Authors:** Aos Aboabat, Zareen Ahmad, Amanda Steiman, Sindhu R. Johnson

**Affiliations:** 1Toronto Scleroderma Program, Mount Sinai Hospital, Toronto Western Hospital, University of Toronto, Toronto, ON M5T 1R8, Canada; 2Toronto Scleroderma Program, Division of Rheumatology, Department of Medicine, Mount Sinai Hospital, University of Toronto, Toronto, ON M5T 1R8, Canada; 3Division of Rheumatology, Department of Medicine, Mount Sinai Hospital, University of Toronto, Toronto, ON M5T 1R8, Canada; 4Toronto Scleroderma Program, Mount Sinai Hospital, Toronto Western Hospital, Institute of Health Policy Management and Evaluation, University of Toronto, Toronto, ON M5T 1R8, Canada

**Keywords:** systemic sclerosis, scleroderma, quality of care, quality improvement, quality indicators

## Abstract

Quality improvement is an emerging field, that applies principles of improvement science and utilizes measurement methods with the aim of improving patient care. Systemic sclerosis (SSc) is a systemic autoimmune rheumatic disease associated with increased healthcare burden, cost, morbidity, and mortality. Gaps in delivering care to patients with SSc have been consistently observed. In this article, we introduce the discipline of quality improvement and its use of quality measures. We summarize and comparatively evaluate three sets of quality measures that have been proposed to evaluate the quality of care of patients with SSc. Finally, we highlight the areas of unmet needs and indicate future directions for quality improvement and quality measures in SSc.

## 1. Introduction

Systemic sclerosis (SSc) is a complex, multi-systemic rheumatic autoimmune disease with associated increased healthcare burden, cost, morbidity, and mortality [1,2]. While it occurs more commonly in women, it is a more aggressive disease in men [3,4,5]. Disease manifestations can vary across ethnicities [6,7] and adversely affects the ability to maintain gainful employment [8,9] and quality of life [10]. Gaps in delivering care to patients with SSc have been consistently observed. One retrospective cohort study showed low numbers of SSc patients who received baseline screening tests and specialty consultations [11]. Another national cross-sectional survey study observed significant variability in the use of diagnostic tests and management among SSc specialty centers [12].

The evaluation of quality in healthcare and the field of quality improvement has rapidly evolved in recent years. The Health and Medicine Division of the National Academies of Sciences, Engineering, and Medicine (previously known as the Institute of Medicine) defines healthcare quality as “the degree to which healthcare services for individuals and populations increase the likelihood of desired health outcomes and are consistent with current professional knowledge” [13]. Multiple North American reports have demonstrated that healthcare quality is suboptimal [13,14,15], thus highlighting the need to improve care quality and prompting efforts to enhance healthcare quality. Quality improvement is a systematic approach of analyzing healthcare performance and the efforts made to improve it [16]. The discipline of quality improvement is underpinned by improvement science theory, and it utilizes measurement methods and tools and applies best practices in its implementation of sustainable quality improvement.

## 2. What Are Quality Measures?

Health quality must be measured to ensure that high-quality care is delivered [17,18]. Quality measures are evidence-based tools designed to help quantify the quality of care, inform systems’ performance, and identify potential healthcare gaps [18,19,20]. This term is often interchangeably used with quality indicators [21]. Quality measures are implemented into clinical practice for many purposes, such as quality improvement initiatives, accreditation, public accountability, and research. They are crucial to all stakeholders at all levels. Physicians may use them to evaluate their practices and implement positive change. Patients can use them to select their providers (in healthcare systems that allow them to) or to evaluate their personal outcomes. Finally, payers and regulators can use them to guide resource allocation and possibly reimbursement [21].

Quality measures are categorized into five major groups: structural measures (innate characteristics of the system and its providers), process measures (what health providers do to people), outcome measures (what happens to people in terms of their health), access, and patient experience [18,20]. A quality measure is composed of: a title; a numerator, which includes the outcome or process of interest during a specified risk period; and a denominator, which includes the population being measured during a reporting period [22]. The definitions and examples of quality measures are summarized in Table 1.

## 3. Quality Measures in Rheumatology

There have been significant shortcomings in delivering care in rheumatology [23,24]. The range of deficits includes increased wait times [25,26,27], suboptimal adherence to quality indicators and guidelines [28,29,30,31,32,33,34], decreased rates of vaccination in immunocompromised patients [35,36], and significant care gaps in youth transitioning from pediatric to adult care [37] across multiple rheumatic conditions. The most crucial and current gold standard step in measuring healthcare quality is through rigorously validated quality indicators that measure processes and outcomes [38]. Quality measures in rheumatology generally focus on process measures as outcomes tend to take years to develop, and they are often influenced by non-quality-related factors (i.e., medical comorbidities, social determinants of health, lifestyle choices, and environmental factors), making them very challenging to accurately measure [39]. Furthermore, if outcome measures are to be used, they must be adjusted for case mix differences (such as the patient’s age or disease severity) to control for their role in influencing the outcome of interest. Such adjustments are needed to ensure that the quality of care delivered is accurately reflected [21].

In efforts that have been made to bridge the quality gap, several quality indicator sets have been developed in many rheumatological conditions, including systemic autoimmune rheumatic diseases (ARDs) [40,41,42,43,44,45], metabolic conditions [46,47], and osteoarthritis [48]. National and international organizations then endorse the quality indicators. They are the basis for developing key performance indicators or performance measures used to quantify care quality for use by professional organizations, governmental, or private entities [23].

The American College of Rheumatology (ACR) published its white paper on quality measures in 2011. Using modified Delphi consensus methods, they defined the essential attributes, priorities, and uses of quality measures. Four areas were identified for the development of future quality measures. These include diseases, medications, comorbidities/prevention, and access/care experience. Rheumatic diseases with the highest priority ranking were rheumatoid arthritis, osteoporosis, juvenile idiopathic arthritis, gout, ankylosing spondylitis, psoriatic arthritis, and osteoarthritis [39]. With the widespread use of electronic health records, the ACR published its first two disease-specific electronic clinical quality measures in rheumatoid arthritis and gout [49,50,51,52], both of which were considered high-priority clinical areas.

Given the significant care gap in rheumatological care and the adoption of care quality as a priority by multiple professional and governmental organizations, there has been an encouraging rise of quality improvement initiatives in rheumatology. Liu et al. summarized quality improvement efforts in adult and pediatric rheumatology from 2013 to 2018. Interventions were focused on improving screening for comorbidities, adherence to clinical practice guidelines, vaccinations, and contraception counselling [53].

## 4. Systemic Sclerosis Quality Measures

The first significant advancement in developing SSc-specific quality measures was in 2011; Khanna et al. developed a set of quality indicators for SSc using consensus methodology [42]. The proposed process indicators were identified based on a literature review and were sent to a group of international SSc experts who refined them. The remaining indicators were evaluated by a United States (US)-based expert panel using the RAND/UCLA appropriateness method. Finally, 32 quality indicators were deemed valid. The final set was presented to the US members of the Scleroderma Clinical Trials Consortium (SCTC), who confirmed its validity and feasibility. The quality indicators were framed in an “IF, THEN, BECAUSE” format across eight disease domains grouped by body systems. The quality indicators were further categorized into 3 different sections: baseline (for patients with a new SSc diagnosis), follow-up monitoring, and treatment (for patients with established SSc), as shown in Table 2.

### 4.1. Baseline

The baseline assessment of a newly diagnosed patient should include antibody tests [54], Doppler echocardiogram, functional status (e.g., Scleroderma Health Assessment Questionnaire [55]), creatinine kinase, and pulmonary function tests, all within the first 12 months. Assessment for tendon friction rubs should occur in the first 3 months, and serum creatinine should be assessed in the first 6 months. If a patient is within the first 5 years of diagnosis, they should be counselled on weekly blood pressure monitoring. If the forced vital capacity or diffusion capacity is less than 80% of the predicted, then a high-resolution CT thorax should be offered within 12 months [42].

### 4.2. Monitoring

Separate quality measures are outlined for the monitoring of a SSc patient with established disease. If the patient experiences new dyspnea on exertion or a new diffusion capacity below 65%, a Doppler echocardiogram should be performed within 3 months. If the examination reveals that proximal muscle weakness and creatine phosphokinase (CPK) is at least three times the normal limit, an electromyogram, muscle biopsy, or magnetic resonance imaging should be performed. For those who have been experiencing symptoms for less than 5 years, spirometry and diffusion capacity should be conducted annually for the first five years. In the case of new dyspnea on exertion, spirometry with diffusion capacity should be conducted within 6 months. If there is interstitial lung disease revealed by chest X-ray, high-resolution CT (HRCT) of the chest, or spirometry; spirometry and diffusion capacity should be performed at least annually until the forced vital capacity (FVC) is stabilized (within 10% over 1 year). If the patient presents new dyspnea on exertion or an abnormal FVC or diffusion capacity of less than 80%, HRCT thorax should be performed within 6 months. If dyspnea on exertion is present and an echocardiogram suggests new pulmonary hypertension, referral for right heart catheterization should be conducted within 3 months. Blood pressure should be recorded during every visit. If hypertension is detected (systolic BP > 140 or diastolic BP > 90 mmHg confirmed on 2 separate occasions), creatinine, complete blood cell count, and urinalysis should be performed within 72 h. Weight or body mass index (BMI) should be recorded annually. Symptoms of gastroesophageal reflux disease should be recorded annually. Hemoglobin test should be offered at least annually [42].

### 4.3. Treatment

Quality measures are also outlined for the treatment of SSc patients. Inactive influenza vaccine should be offered annually, while the pneumococcal vaccine should be offered every 5 years, unless contraindicated. If symptoms of diastolic dysfunction and heart failure are present, appropriate treatment such as ACE inhibitors, diuretics, or beta-blockers should be provided within three months, or a referral to a cardiologist should be performed. For patients with NYHA/WHO functional class II–IV secondary to pulmonary arterial hypertension, as diagnosed through right heart catheterization, treatment with endothelin blockers, prostacyclin analogs, and/or PDE-5 inhibitors should be initiated within three months. For those with a decreased range of motion or function in the hand and a diagnosis of less than five years, a range-of-motion exercise program should be offered within 6 months. In cases of interstitial lung disease and a greater than 10% decline in FVC over the past 12 months, immunosuppressive treatment options such as cyclophosphamide, methotrexate, azathioprine, cyclosporine, or mycophenolate mofetil should be offered within 3 months. If a patient is experiencing scleroderma renal crisis (accelerated hypertension [at least SBP ≥ 140 and a rise of SBP ≥ 30 mmHg from baseline] or rapidly progressive renal failure), an ACE inhibitor should be prescribed within 72 h. A proton pump inhibitor or H2 blocker should be offered for patients diagnosed with GERD within 3 months of diagnosis. Symptoms of early satiety, post-prandial abdominal bloating, postprandial vomiting, or regurgitation persisting for at least 1 month should prompt testing for impaired gastric emptying or an empiric trial of therapy within 6 months. If a patient experiences unintentional weight loss of 5% or more over 3 months with symptoms of nausea, vomiting, bloating, or diarrhea for 4 weeks, testing for malabsorption or bacterial overgrowth or an empiric trial of therapy should be offered within 3 months. If a patient has digital tip ulcers, treatment with calcium channel blockers, prostacyclin therapy, topical nitrate therapy, or PDE-5 inhibitors should be prescribed within 3 months [42].

Hoffmann-Vold et al. developed tools for the annual assessment of patients with SSc to facilitate an international standardization of follow-up care [56]. Using a stepwise Delphi consensus method, 157 multidisciplinary SSc expert and non-expert physicians from the European Scleroderma Trials and Research group (EUSTAR) and SCTC rated the proposed disease domains and assessment tools by the study authors. The domains and tools were included in the final quality indicator set if they were rated higher than 80% in importance by more than 75% of participants. Ten domains were deemed important by consensus: Raynaud’s phenomenon, digital ulcers, mucocutaneous, musculoskeletal, lung, heart, gastrointestinal, renal, laboratory, and treatment. The specific symptoms and assessment tools of the domains are listed in Table 3. The investigators felt these quality indicators could be easily applied across worldwide healthcare systems, including non-academic centers.

Spierings et al. surveyed 650 SSc patients across 13 hospitals in the Netherlands using an online questionnaire [57]. Patients were asked to rank the proposed process, and outcome quality indicators developed following focus group interviews with patients, rheumatologists, and specialized nurses. The proposed process indicators were a good physician–patient relationship, multidisciplinary collaboration, receiving guideline-directed therapy, annual evaluations of pulmonary function and skin involvement, healthcare access, counselling of non-pharmacological care, and accuracy of diagnosis. The outcome indicators suggested were the absence of organ involvement, lack of disease progression, absence of digital ulcers, improved quality of life, reduced pain, reduced fatigue, and improved hand function. Among the proposed process indicators, the top three rated indicators were: the physician–patient relationship, multidisciplinary collaboration, and receiving guideline-directed therapy. Outcome indicators that were considered most important were the absence of disease progression, the absence of organ involvement, and the absence of digital ulcers, Table 4.

## 5. Comparative Evaluation of SSc Quality Measures

Choosing how to measure the quality of care in SSc is a challenge. Different stakeholders (patients, physicians, nurses, allied health staff, governments, and policymakers) can have differing priorities and perspectives. Indeed, there appear to be differences in quality measures between patients and physicians. SSc experts prioritize disease-specific processes as quality measures. This is also compatible with some process indicators identified by SSc patients, such as annual lung assessments and receiving appropriate therapy.

However, patients value processes that are not specific to their disease, such as their relationship with their providers and multidisciplinary collaboration. To date, healthcare providers and researchers have not established SSc outcome indicators. However, disease outcomes are a high priority for patients. There are also differences between physician-derived quality indicator sets. Compared to the quality indicator set of Khanna et al., Hoffman et al. additionally advocate for the inclusion of indicators related to Raynaud’s phenomenon (documentation of severity and frequency of attacks), digital ulceration (new ulcer occurring in the past year), co-morbidities affecting perfusion (i.e., diabetes mellitus), smoking status, cutaneous disease (telangiectasias, calcinosis), musculoskeletal disease (arthritis), and comorbidity (heart disease).

## 6. Unmet Needs and Future Directions

Although there has been remarkable progress made in the quality improvement field in SSc over the last decade, much remains to be done. Rigorously developed quality measures that reflect stakeholders’ priorities are essential for measuring the quality of care delivered to patients with SSc. It is uncertain if these can be successfully adhered to. Many of the quality measures recommended for the baseline assessment of SSc as they are required for the classification of SSc [58]. Many of the quality measures recommended for the monitoring of SSc patients with established disease are implemented worldwide [59]. However, it may be challenging to adhere to so many quality indicators. If the number of quality indicators is to be reduced to a more manageable number, it should be questioned: whose perspective should be prioritized? [60,61] (Patients? Physicians? Payers?) One shared provider–patient approach is the use of a health passport, which outlines quality measures and their timing [62]. This passport could serve as a point of discussion or reminder for a quality measure to occur.

Furthermore, it is essential that quality measures for systemic sclerosis are regularly updated to reflect recent advances in the field. Examples of such advances include the 2013 ACR/EULAR classification criteria [63], the 2018 ACR/EULAR treatment guidelines [64], and the validation of multiple SSc-specific patient-reported outcomes (PROs) [65,66,67] to ensure that patients receive the most appropriate and effective care using the most current and relevant guidelines.

To date, there are no quality improvement initiatives addressing care deficiencies in SSc. Local quality improvement initiatives are needed to measure SSc quality indicators uptake and diagnose any potential deficiencies. If a quality gap is established, then rapid quality improvement cycles can be designed to address it. Rapid improvement cycles use quality improvement methodology to develop a timely and specific aim, choosing a family of process, balancing and outcome measures, and designing thoughtful interventions to address the causes of the targeted gap. Interventions are then regularly defined using plan–do–study–act (PDSA) cycles to evaluate their efficacy until the target gap is bridged [68], as shown in Figure 1.

## 7. Conclusions

This discipline of quality improvement to improve the quality of care is rapidly evolving in rheumatology and emerging in SSc. Three sets of SSc quality indicators have been proposed. The physician-derived quality indicators are limited to process indicators. SSc quality indicators can be categorized by body system and/or function (screening, monitoring, and treatment). While there is some overlap between patient-derived and physician-derived quality indicators, patients also value outcome quality indicators. It is important for physicians to remember that patients prioritize quality indicators differently, giving higher priority to non-SSc-related quality indicators.

## Figures and Tables

**Figure 1 diagnostics-13-00579-f001:**
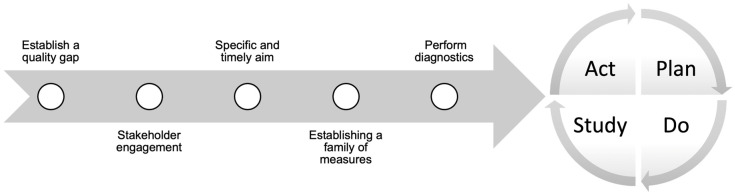
Quality improvement process.

**Table 1 diagnostics-13-00579-t001:** Healthcare quality measure domains and systemic sclerosis-related examples.

Domain	Definition	SSc-Related Example
Structural	A characteristic of a healthcare organization or individual practitioner that pertains to their ability to deliver high-quality healthcare.	Number of rheumatology clinics in which the modified Rodnan skin score is documented.
Process	A healthcare process refers to an action or series of actions taken for, on behalf of, or by a patient in the course of their care.	Liver function test measurement after mycophenolate therapy.
Outcome	A health state of a patient resulting from healthcare.	A proportion of SSc patients with inflammatory arthritis achieving low disease activity after starting DMARD or biologic therapy
Access	Access to healthcare is the ability of patients or enrollees of a healthcare organization or individual practitioner to receive timely and appropriate medical care.	Percentage of SSc patients who see a rheumatologist within 3 months of symptom onset in a particular geographic area.
Patient Experience	Patients’ accounts of their healthcare experience and its impact on their health.	Patients’ satisfaction following the rheumatology encounter.

DMARD, disease modifying anti-rheumatic disease therapy.

**Table 2 diagnostics-13-00579-t002:** Systemic sclerosis process of care quality indicators derived from physicians for use in the US healthcare system.

Domain	Baseline Assessment in Newly Diagnosed Patients	Monitoring of Established SSc	Treatment
General	Tests for Topoisomerase I, centromere, and RNA polymerase III antibodies should be performed within 12 months.	Hemoglobin test should be offered at least annually	Inactive influenza vaccine should be offered annually unless contraindicated
			Pneumococcal vaccine should be offered every 5 years unless contraindicated
Cardio-Pulmonary	Doppler echocardiogram within 12 months	If new onset dyspnea on exertion or a decline in DLCO to <65% predicted, a doppler echocardiogram should be performed within 3 months	If symptoms of diastolic dysfunction and symptomatic heart failure, then treatment (e.g., ACE inhibitor, diuretic, beta-blocker) or a referral to a cardiologist should be offered within 3 months
			For patients with NYHA/WHO functional class II-IV secondary to pulmonary arterial hypertension, as diagnosed through right heart catheterization then treatment (endothelin blockers, prostacyclin analogs, and/or PDE-5 inhibitors), should be initiated within 3 months
Physical function	Assessment of functional status (e.g., activities of daily living, health assessment questionnaire-disability index, or self-reported measures) should be conducted within one year.		
Musculoskeletal	Serum creatine phosphokinase (CPK) within 12 months	If examination reveals proximal muscle weakness and creatine phosphokinase (CPK) is ≥3 times the normal limit, an electromyogram, muscle biopsy, or magnetic resonance imaging should be performed	If first signs or symptoms <5 years and has decreased hand function or range of motion, a range-of-motion exercise program should be offered within six months
	If palpable tendon friction rub(s) present, a follow-up visit should be offered within 3 months		
Pulmonary	Spirometry and DLCO should be offered within 12 months	If <5 years from first signs or symptoms, then spirometry and DLCO should be offered at least annually for the first 5 years	If interstitial lung disease present and >10% decline in FVC over the past 12 months, immunosuppressive treatment options such as cyclophosphamide, methotrexate, azathioprine, cyclosporine, or mycophenolate mofetil should be offered within 3 months
		If new dyspnea on exertion, spirometry with DLCO should be performed within 6 months	
		If interstitial lung disease revealed by chest X-ray, high-resolution CT (HRCT) of the chest or spirometry, spirometry and DLCO should be performed at least annually until the FVC is stabilized (within 10% over 1 year)	
		If new dyspnea on exertion or an abnormal FVC or DLCO <80% predicted, HRCT thorax should be performed within 6 months	
		If new dyspnea on exertion is present and an echocardiogram suggests new pulmonary hypertension (estimated right ventricular systolic pressure >50 mm Hg or tricuspid regurgitation velocity >3.5 mm/s), referral for right heart catheterization should be conducted within 3 months	
	If FVC or DLCO <80% predicted, then a high-resolution CT thorax should be offered within 12 months		
Renal	Serum creatinine should be offered within 6 months	Document a blood pressure measurement at every clinic visit	If scleroderma renal crisis (accelerated hypertension [at least SBP ≥140 and a rise of SBP ≥30 mmHg from baseline] or rapidly progressive renal failure), then prescribe an ACE inhibitor within 72 h
	If <5 years from first signs or symptoms, document counseling to perform at least weekly blood pressure measurements		
		If new onset hypertension (systolic BP >140 or diastolic BP > 90 mmHg confirmed on 2 separate occasions), then serum creatinine, CBC with platelets, and urinalysis should be offered within 72 h	
Gastrointestinal		Document weight or body mass index at least annually	A proton pump inhibitor or H2 blocker should be offered for patients diagnosed with GERD within 3 months of diagnosis.
		Symptoms of GERD (e.g., heartburn, nocturnal cough, dysphonia, acid taste, chest pain) should be recorded annually	Symptoms of early satiety, post-prandial abdominal bloating, postprandial vomiting, or regurgitation persisting for at least 1 month should prompt testing for impaired gastric emptying (e.g., upper endoscopy, gastric emptying study, upper GI series) or an empiric trial of therapy (e.g., prokinetics, PPI) within 6 months
			If unintentional weight loss ≥ 5% over 3 months with symptoms of nausea, vomiting, bloating, or diarrhea for 4 weeks, testing for malabsorption or bacterial overgrowth (e.g., lactulose breath test, glucose breath test, xylose test, jejunal culture, serum carotene, fecal fat determination) or an empiric trial of therapy (e.g., antibiotics, prokinetics, octreotide) should be offered within 3 months
Peripheral vascular			If digital tip ulcer(s) develop, therapy (e.g., calcium channel blockers, prostacyclin therapy, topical nitrate therapy, PDE-5 inhibitor) should be initiated within 3 months

DLCO: diffusing capacity of the lungs for carbon monoxide; ACE: angiotensin-converting enzyme; NYHA: New York Heart Association; WHO: World Health Organization; PDE-5: phosphodiesterase-5; FVC: forced vital capacity; HRCT: High-resolution computed tomography; CT: computed tomography; SBP: systolic blood pressure; CBC: complete blood count; GERD: gastro-esophageal reflux disease; GI: gastrointestinal. Adapted from Khanna et al., 2011 [42].

**Table 3 diagnostics-13-00579-t003:** Annual assessment of organ involvement in systemic sclerosis.

Domain	Symptoms	Assessment Tools
Raynaud’s phenomenon	Frequency and severity of attacks	
Digital ulcers	Fingertip and proximal (of DIP joints) ulcers, development of new ulcers during the past year, underlying conditions that may affect perfusion (i.e., diabetes), and smoking status	
Mucocutaneous	Skin changes (worsening or improvement; patient-reported)	Puffy fingers Modified Rodnan Skin Score Telangiectasias Calcinosis
Musculoskeletal	Muscle weakness and stiffness	Puffy fingers Joint contractures Arthritis Calcinosis Tendon friction rub count
Lung	Dyspnea	Functional class (NYHA 1–4) Lung crackles at the bases on auscultation Lung function test and DLCO
Heart	Dyspnea	Functional class (NYHA 1–4) Leg edema Electrocardiogram Doppler-echocardiography Heart rate Blood pressure Concurrent heart disease
Gastrointestinal	Night and daytime heartburn/reflux, dysphagia, diarrhea, and weight loss	Weight
Renal		Serum creatinine Estimated glomerular filtration rate Urine analysis Blood pressure
Laboratory		Acute phase reactants, creatine kinases, hematology, renal function test, liver function tests
Treatment		Type of treatment (generic and name of drug) Date of initiation Date of finalization

DIP: distal interphalangeal; NYHA: New York Heart Association; DLCO: diffusing capacity of the lungs for carbon monoxide. Adapted from Hoffmann-Vold et al., 2019 [56].

**Table 4 diagnostics-13-00579-t004:** Process and outcome quality indicators ranked by SSc patients.

Ranking	Process Quality Indicators	Outcome Quality Indicators
1	Good physician–patient relationship	Absence of organ involvement
2	Multidisciplinary collaboration	Absence of disease progression
3	Receiving guideline-directed therapy	Absence of digital ulcers
4	Annual evaluations of pulmonary function	Improved quality of life
5	Annual evaluations of skin scores	Pain reduction
6	Healthcare access	Fatigue reduction
7	Counselling of non-pharmacological care	Improved hand function
8	Accuracy of diagnosis	

Adapted from Spierings J, et al., 2020 [57].

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
