# Peer review of "Quality Measures in Systemic Sclerosis"

_diagnostics, 2023, doi:10.3390/diagnostics13040579_

Round 1

Reviewer 1 Report

This is a nice and under-considered topic in SSc, as physician frequently focus on providing care and treatment, without considering the quality. 

Despite the high need of quality measures and check, the authors should highlight not only the discrepancy between patient and physician vision, but also the need for the measures to be regularly updated. Most of the Khanna 2011 are not relevant anymore, given the progress and information accumulated in the past decade. 

Moreover, is there any datum regarding the implementation of the QM? Any feasibility assessment, or evaluation of how many centers could offer what is considered to be a quality measure?

Some minor comment:

- page 6, lines 158-160. Is the subject missing? This sentence is unclear to me.

- tables overall: when you state "not applicable", do you refer to an "empty cell", as some domain has 1 item and other 2-3? It is a bit misleading, I initially thought that some QM was considered as "not applicable". I would maybe adjust the tables, just leaving empty cells. 

Author Response

Dear reviewers,

We appreciate the time and effort that you have taken to evaluate our manuscript. We have carefully considered all the comments and have revised the manuscript to address the concerns raised. In this letter, we will summarize the changes that have been made in response to the reviewers' feedback.

Response to Reviewer 1 Comments:

Point 1: This is a nice and under-considered topic in SSc, as physician frequently focus on providing care and treatment, without considering the quality. Despite the high need of quality measures and check, the authors should highlight not only the discrepancy between patient and physician vision, but also the need for the measures to be regularly updated. Most of the Khanna 2011 are not relevant anymore, given the progress and information accumulated in the past decade.

Response 1: We have added a paragraph to address this under the (Unmet Needs and Future Directions): “Furthermore, it is essential that quality measures for systemic sclerosis are regularly updated to reflect recent advances in the field. Examples of such advances include the 2013 ACR/EULAR classification criteria, the 2018 ACR/EULAR treatment guidelines, and the validation of multiple SSc-specific patient-reported outcomes (PROs) to ensure that patients receive the most appropriate and effective care, using the most current and relevant guidelines”.

Point 2: Moreover, is there any datum regarding the implementation of the QM? Any feasibility assessment, or evaluation of how many centers could offer what is considered to be a quality measure?

Response 2: To the best of our knowledge, there has been no previous data on the uptake and implementation of these QM in clinical practice. We are currently assessing the uptake and developing interventions to address any gaps in our center, in order to ensure the delivery of high-quality care to our patients. We plan to publish the results of our efforts. We have added to the manuscript: “Finally, there are no data regarding the implementation of quality measures in the care of people with systemic sclerosis. Feasibility assessments, or evaluation of how many centers could offer what are considered to be a quality measures are needed across practice settings and jurisdictions.”

Point 3: - page 6, lines 158-160. Is the subject missing? This sentence is unclear to me.

Response 3: This was corrected to “For those with decreased range of motion or function in the hand and a diagnosis of less than five years, a range-of-motion exercise program should be offered within six months”.

Point 4: - tables overall: when you state "not applicable", do you refer to an "empty cell", as some domain has 1 item and other 2-3? It is a bit misleading, I initially thought that some QM was considered as "not applicable". I would maybe adjust the tables, just leaving empty cells.

Response 4: By "not applicable," we mean an empty cell. We have revised the design accordingly.

Response to Reviewer 2 Comments:

Point 1: Page 3 line 99: please define QI;

Response 1: This was defined = Quality improvement.

Point 2: Page 3-5 Table 2: please add footnotes with all abbreviation used in Table 2;

Response 2: These were added.

Point 3: Page 6 line 125: please add the abbreviation for high resolution CT (should be there and not in line 140);

Response 3: We paraphrased this paragraph and added the abbreviation.

Point 4:  Page 6 line 131: please define CPK;

Response 4: This was defined.

Point 5: Page 6 line 147: please correct the typos "((";

Response 5: We paraphrased this paragraph and corrected the typos.

Point 6:  Page 7-8 Table 3: please add footnotes with all abbreviation used in Table 3 (i.e. DIP was not defined).

Response 6: These were added.

Thank you for the thoughtful reviews and the opportunity to resubmit this manuscript.

Sincerely,

Aos Aboabat MBBS

Sindhu Johnson MD PhD

Reviewer 2 Report

The Authors provided an interesting overview about quality measures that have been proposed to evaluate the quality of care of patients with systemic sclerosis (SSc). Moreover, they highlighted the importance of local quality improvement initiatives to measure SSc quality indicators to address deficiencies. The manuscript is well written and organized, however there are some concerns to be addressed:

1) Page 3 line 99: please define QI;

2) Page 3-5 Table 2: please add footnotes with all abbreviation used in Table 2;

3) Page 6 line 125: please add the abbreviation for high resolution CT (should be there and not in line 140);

4) Page 6 line 131: please define CPK;

5) Page 6 line 147: please correct the typos "((";

6) Page 7-8 Table 3: please add footnotes with all abbreviation used in Table 3 (i.e. DIP was not defined).

Author Response

(The authors gave the same response as above.)
